# VERSATILE OUTLIER DETECTION WITH OUTLIER PRE-SERVING DISTRIBUTION MAPPING AUTOENCODERS

## ABSTRACT

State-of-the-art deep learning methods for outlier detection make the assumption that outliers will appear far away from inlier data in the latent space produced by distribution mapping deep networks. However, this assumption fails in practice, because the divergence penalty adopted for this purpose encourages mapping outliers into the same high-probability regions as inliers. To overcome this shortcoming, we introduce a novel deep learning outlier detection method, called Outlier Preserving Distribution Mapping Autoencoder (OP-DMA), which succeeds to map outliers to low probability regions in the latent space of an autoencoder. For this we leverage the insight that outliers are likely to have a higher reconstruction error than inliers. We thus achieve outlier-preserving distribution mapping through weighting the reconstruction error of individual points by the value of a multivariate Gaussian probability density function evaluated at those points. This weighting implies that outliers will result in an overall penalty if they are mapped to low-probability regions. We show that if the global minimum of our newly proposed loss function is achieved, then our OP-DMA maps inliers to regions with a Mahalanobis distance less than $\delta$, and outliers to regions past this $\delta$, $\delta$ being the inverse Chi Squared CDF evaluated at $1 - \alpha$ with $\alpha$ the percentage of outliers in the dataset. Our experiments confirm that OP-DMA consistently outperforms the state-of-art methods on a rich variety of outlier detection benchmark datasets.

## 1 INTRODUCTION

**Background and Motivation.** Outlier detection, the task of discovering rare or abnormal instances in a dataset, is critical for many applications from fraud detection, error identification in measurements, to fault detection in systems (2). Which points are outliers is often unknown in practice, and thus labeling for which points are outliers versus inliers is scarce or even completely unavailable. For this reason, in this work we focus on *unsupervised methods for outlier detection* (1).

In the ideal case in which inliers follow a simple unimodal distribution and outliers occur "far away" from the mean compared to the inliers, outlier detection has a simple solution. In this special case, outliers can be found by simply hardcoding some fixed distance as the cutoff between which points are classified as inlier or as outlier, such as is done by EllipticEnvelope (7). Unfortunately, in practice data is often not distributed in such a convenient manner (13). Rather, inliers are distributed in a complex, unknown fashion potentially with multiple clusters of inliers in the feature space (6).

Other methods with not quite as strict assumptions on the distribution of the data can work on arbitrary distributions, such as One-Class methods (4) and IsolationForest (3) to name just a few. These methods typically use a density-based approach (3; 4; 23). However, these methods are known to become infeasible in high dimensional data sets, because in sufficiently high dimensional spaces, all data points are known to be roughly equidistant and the space around all points will be sparse (5).

**State-of-Art Deep Learning Methods for Outlier Detection.** Deep learning methods for outlier detection not only overcome the limitations of density and distance based methods, but they can also leverage the inherent featurization capabilities of deep networks. Recently, deep learning methods use the reconstruction error of an autoencoder-based model for outlier detection, either by considering the reconstruction error itself as an outlier score (18; 20) or by leveraging the reconstruction error in a more sophisticated way (10; 19; 9). However, it has been shown that the usage of the mere reconstruction error can be a suboptimal choice for an outlier score because outliers tend to converge

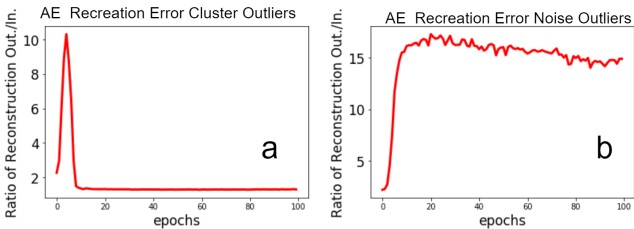

Figure 1: Ratio of average reconstruction error of outliers/inliers for an autoencoder trained on Gaussian data with cluster outliers (a) or uniform noise outliers (b)

to an average reconstruction error that is indistinguishable from that of inliers as the network converges (19). This is a greater issue for the case where outliers occur in clusters as opposed to uniform noise, as shown in Figure 1. The cluster outlier case is is likely to arise when the outliers represent a small minority class or when they are caused by a systematic error.

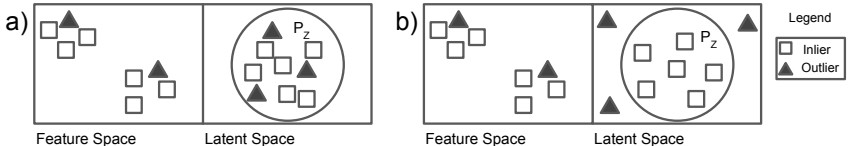

Figure 2: a) Standard deep distribution mapping methods that enforce a certain prior distribution on the latent space (VAEs, WAEs) in practice map both outliers and inliers to high probability regions. b) The erroneous assumption of many methods is that outliers are mapped to regions far from the high probability region of the prior distribution.

To go beyond this problem of the pure reconstruction error, recent works have leveraged the distribution-mapping capabilities of generative models such as Variational Autoencdoers (VAEs) that encourage the data in the latent space to follow a prior distribution (19; 21). These methods assume that due to their anomalous nature outlier points will not be mapped to the encouraged prior distribution while the inliers will be, and consequently outliers will occur in low-probability regions of the prior distribution (19; 21) [Figure 2 (b)]. However, it was observed that this widely held assumption is faulty in practice (21), with the distribution mapping continuing to map both inliers and outliers to high probability regions of the prior distribution for a wide variety of applications [Figure 2 (a)].

**OP-DMA: Our Approach.** In this work, we propose a method which leverages the distribution mapping capabilities of *generative models*, but unlike other distribution mapping outlier detection methods we no longer conveniently assume that outliers are not well mapped to the enforced prior distribution. Our core idea instead is to design a novel prior probability-weighted loss function that actively encourages outliers to be mapped to low probability regions in the latent space. Our method is an autoencoder architecture that we henceforth refer to as the Outlier Preserving Distribution Mapping Autoencoder (OP-DMA). The network is trained to minimize both the Maximum Mean Discrepancy (MMD) distance between the distribution of the data in the latent space and a multivariate Gaussian distribution, and what we call a prior-weighted L2 distance between the input and the reconstruction of the input. This prior-weighted distance weights each input datapoint point by the corresponding value of the multivariate Gaussian PDF evaluated at that point's latent representation. The prior-weighted L2 distance encourages points with a high reconstruction error, as we assume outliers have, to be placed in low-probability regions in the latent space. This corresponds to the important observation that outlier will be mapped far from the mean of the data. The beauty of this mapping is that it thus allows us to use a simple off-the-shelf distance-based metric to identify outliers post-transformation. As the original distribution of the data in the feature space is arbitrary and no distance or density assumptions are placed on the data, OP-DMA thus corresponds to a highly versatile outlier detection strategy. Our key contributions are summarized as follows:

1. Introduce OP-DMA, a novel distribution-mapping autoencoder that preserves ourliers in the latent space without making asumptions on the distribution of the data in the feature space.

2. Prove that the optimal solution for OP-DMA places outliers further than $\sqrt{\int_0^{1-\alpha} \frac{t^{-n/2-1}e^{\frac{1}{2t}}}{2^{\frac{n}{2}}\Gamma(\frac{n}{2})}\,dt}$ according to the Mahalanobis distance, where $\alpha$ is the probability that draw from a given dataset is an outlier. This allows us to use a distance-based outlier detection classifier

3. Show experimentally that OP-DMA routinely outperforms standard distribution-mapping autoencoders for outlier detection on a variety of real-world benchmark datasets

## 2 RELATED WORK

**Outlier Detection Using Autoencoders Coupled with Classical Outlier Detectors.** Erfani et al. (11) used autoencoders for dimension reduction before classifying the projected data with a One Class SVM. Raghavendra et al. (12) introduce a *One Class Neural Network* (OC-NN), a feed-forward neural network inspired by One Class SVMs for the outlier-detection. The OC-NN method first pretrains an autoencoder before introducing a OC-NN layer after the encoder. Unlike our method (OP-DMA) these methods do not encourage outliers in the feature space to remain outliers in the latent space, making the strong and in practice often incorrect assumption that outliers in the feature space remain outliers in the latent space.

**Recreation Error Methods.** Zhou and Paffenroth (10) introduced Robust Deep Autoencoders (RDA), a deep network version of Robust Principal Component Analysis. It separates a given data matrix $D$ into a low-rank matrix $L$ and a sparse matrix $S$, where anomalous datapoints are isolated in $S$. This forces the network to reconstruct the instances in $L$, and punishes the network for putting instances into $S$. Chen et al. (9) introduced *RandNet*, an ensemble of autoencoders for outlier detection with outlier score corresponding to the average recreation error among the networks. Sabokrou et. al. introduced *Adversarially Learned One-Class Classifiers (ALOCC) for Novelty Detection* (27). ALOCC consists of a generator network $\mathcal{R}$ which is trained to reconstruct noise-contaminated images by adversarially training $\mathcal{R}$ with $\mathcal{D}$, a discriminator network which distinguishes between real images and reconstructed images. They show that the reconstruction error for $\mathcal{R}$ as well as the output of discriminator $\mathcal{D}$ are higher for outliers as opposed to inliers. Xia et. al. (28) used an autoencoder for outlier detection by adaptively grouping unlabeled input data into either an *outlier* or *normal* class, such that the network is trained to minimimize variance of reconstruction error within a class while increasing variance overall. However, as shown in (19), for autoencoders the reconstruction error of outliers often converges to that of inliers. This negatively impacts the performance of such reconstruction error methods.

**Generative Outlier Detection Methods.** Perera et. al. (21) proposed adversarial autoencoders with two discriminator networks for outlier detection. The first discriminator encourages the representation of the data in the latent space to be uniform, while the second encourages the output of the decoder conditioned on random samples from the uniform distribution to match real data samples. Vu et. al. (20) used an adversarial autoencoder to map the latent representation of their data to a standard normal distribution. They then combine the outlier score from a One Class SVM on the latent data with the One Class score on the reconstructed data as the final outlier score. Liu et. al. proposed MO-GAAL (26), a deep generative model which uses multiple generator networks to create outlier datapoints near subsets of the input data. A dsicriminator network is used to identify boundaries around the subsets of real data such that the generated outliers are rejected. These distribution mapping methods rely on the assumption that unlike the inliers, the outliers will be mapped to low-orobability regions of the prior distribution. However, as we discussed in Section 1, in practice both inliers and outliers are often well mapped to the prior distribution. This is because outliers that are mapped to low-probability regions will generally incur a higher cost from the divergence term which matches the latent distribution to the prior, and unlike in our OP-DMA approach there is not a corresponding term to counteract this cost.

## 3 PROPOSED METHOD: OUTLIER PRESERVING DISTRIBUTION MAPPING

The goal of OP-DMA is to find a latent representation of a dataset X such that outlier in X can be easily isolated from inliers in the latent representation of X. That is, we find a transformation $f$ that we can apply to a given dataset $X$ with a distribution $P_X$ such that $f(X) \sim P_Z$, where $P_X$ is an arbitrary

and unknown distribution of our input dataset $X$ and $P_Z$ is the target distribution. Further, if $a \in X$ is an outlier, then $f(a)$ should be an outlier in $f(X)$ as well. That is, $p_Z(f(a))$ should be small. For the purpose of outlier detection, we chose $P_Z$ to be a Gaussian distribution as low probability regions would then simply be values far from the mean. We chose an autoencoder architecture to accomplish this distribution mapping, where $f$ is selected to be the encoder $Q$ parameterized by weights $\theta$ that are learned through stochastic backpropagation.

OP-DMA accomplishes the goal of mapping outliers to low probability regions on $P_Z$ by weighing the autoencoder's reconstruction error by the likelihood of the latent data in the latent space with distribution $P_Z$. In order to accomplish this, we need a one-to-one correspondence between each original datapoint, the latent representation of that datapoint, and the reconstructed point. Additionally, while we want the distribution of the latent data to follow $P_Z$ on the whole, we do not want all points to be mapped to high-probability regions of the prior. Rather, we must develop a solution that encourages some datapoints (i.e. outliers) to have a low likelihood in the prior distribution. For these reasons, a Wasserstein Autoencoder (WAE) is a more appropriate choice than the more common Variational Autoencoder (VAE). We chose WAEs, instead of the more commonly used VAEs, because the WAEs encourages the latent representations as a whole to match the prior, whereas the loss function of a VAE encourages each individual latent point to represent a distribution that matches the prior (15). This is counter to the goal of OP-DMA, which is to have some points (i.e., the outliers) mapped away from the prior distribution. Additionally, by formulating the reconstruction process as a Wassertein distance we can leverage the distribution mapping capabilities of the generative model while still operating with both a deterministic encoder and decoder. This is important as this allows us to maintain a one-to-one correspondence between a given point in the original space and its corresponding mapped point in the latent space so we can weight the reconstruction error by the value of the prior PDF evaluated in the latent space.

## 3.1 WASSERSTEIN AUTOENCODERS

As OP-DMA is an extension of WAE, we first briefly describe the structure of WAEs. WAEs are distribution-mapping autoencoders which minimize the Wasserstein distance between original data and its reconstruction. For original data X and reconstruction Y, it is defined as:

$$W_c(P_X, P_Y) := \inf_{\gamma \in \Gamma} \int \int c(x, y)\gamma(x, y)dxdy \qquad (1)$$

where $\Gamma$ denotes the set of all joint distributions of $X$ and $Y$, such that the marginal distributions $\Gamma(X|Y)$ and $\Gamma(Y|X)$ are distributed according to $P_X$ and $P_Y$ respectively, and $c$ is a Wasserstein-divergence cost function. That is, the c-Wasserstein distance is the expectation of the cost function $c$ taken with respect to the joint distribution $\gamma$ with marginals equal to the two priors, such that $\gamma$ is a joint distribution that produces the minimum expectation of Equation 1.

Simply finding the Wasserstein distance, which is the distance that a WAE is trained to minimize, is itself an optimization problem. However, it is shown in (17) that when the encoded data $Q(X) \sim P_Z$, where $Q$ is the encoder network and $P_Z$ is the prior, then the search over all joint distributions previously required to compute the Wasserstein distance can be replaced with a search over all random encoders, $\inf_{\gamma \in \Gamma} \mathbb{E}_{(X,Y) \sim \gamma}[c(X, G(Z))] = \inf_{Q:P_Q=P_Z} \mathbb{E}_{P_X} \mathbb{E}_{Q(Z|X)}[c(X, G(Z))]$. This means we can minimize the c-Wasserstein distance by finding the weights of the encoder that produce the minimum expectation of cost $c$. As is done in (16) where WAE is defined, we can relax the constraint that $P_Q \sim P_Z$ by adding a penalty term $\mathcal{D}(P_Q, P_Z)$, where $\mathcal{D}$ is some function that measures the divergence between $P_Q$ and $P_Z$. Thus,

$$W_c^\lambda(X, Y) = \inf_Q \mathbb{E}_{P_X} \mathbb{E}_{Q(Z|X)}[c(X, G(Z))] + \lambda \mathcal{D}(P_Q, P_Z), \qquad (2)$$

where $\lambda$ is a constant weight term that determines how much the divergence of $P_Q$ from the prior $P_Z$ is penalized. In the case of the optimal solution for $Q$, $W_c^\lambda(X, Y) = W_c(X, Y)$. In that case the Wasserstein objective will equal the true c-Wasserstein distance. While the WAE objective function encourages the distribution of the latent data as a whole to match the prior distribution, WAEs do not encourage outliers in the feature space to remain outliers in the latent space. Even though the

divergence term does not explicitly encourage every point to occur in a high probability region, in practice this is what happens. Consider when $\mathcal{D}$ is a discriminator network. The discriminator is likely to learn a boundary around the high probably region of the prior distribution, and the encoder network will be penalized for mapping an outlier to a low probablity region outside of the boundary as the discriminator would correctly identify it as a generated point. On the other hand, as described below, our proposed Outlier-Perserving Distribution Mapping Autoencoders actively encourages outliers to be mapped to low probability region in the latent space.

## 3.2 Outlier-Preserving Distribution Mapping Autoencoders (OP-DMA)

So far, we have formulated the distribution mapping process in terms of random encoders and decoders $Q$ and $G$. However, we can deterministically produce $Q(X)$ and $G(Q(X)|X)$ by requiring $Q(X) = \delta_{\mu(X)}$, where $\mu$ is some function mapping $X$ to $Q(X)$. This allows us to have a one-to-one correspondence between input points $X$ and output $G(Q(X))$ points, so that $\mathbb{E}_{P_X}\mathbb{E}_{Q(X)}[c(X, G(Q(X)))]$ can define a proper reconstruction error between each point $x \in X$ and its corresponding output $y \in Y$. This reconstruction error can be used to determine which points should be mapped to low probability regions in the latent space. Let us define $c' := c(x, G(Q(x))) \cdot p_Z(Q(X))$. We can thus define the OP-DMA objective function, $W_{c'}^{\lambda}$, as:

$$W_{c'}^{\lambda} = \inf_{Q:P_Q=P_Z} \mathbb{E}_{P_X}\mathbb{E}_{Q(Z|X)}[c'(X, G(Z))] + \lambda\mathcal{D}(P_Q, P_Z) \tag{3}$$

We can think of $p_Z(Q(X))$ as a weight on the reconstruction term. Thus, high-reconstruction error points that are mapped to high-probability regions will be penalized more than high-reconstruction error points which are mapped to low probability regions. Since outliers are assumed to result in a high reconstruction error (at least during early epochs of training), by lessening the penalty to the network for poorly reconstructed points that have been mapped to low-probability regions of the prior we encourage the network to map outlier to these low-probability regions. However, the encoder isn't encouraged to map every point (that is, both inliers and outliers) to low probability regions, as the $\mathcal{D}$ term makes it so that the distribution of all the encoded points on the whole should well match the prior.

Although we have made a significant modification to the Wasserstein loss function, as Theorem 1 states minimizing this cost function also corresponds to minimizing a lower bound on the non-prior-weighted Wasserstein divergence:. **The proofs of all theorems below are provided in the appendix for space reasons.**

**Theorem 1.** *Let $P_Z = \mathcal{N}(\mu, \Sigma)$. Then $W_{c'}(X, G(Q(X))) \leq W_c(X, G(Q(X)))\frac{1}{(2\pi^{n/2})|\Sigma|^{1/2}}$. Additionally, if $|\Sigma| \leq \frac{1}{(4\pi^n)}$ then $W_{c'}^{\lambda}(X, G(Q(X))) \leq W_c^{\lambda}(X, G(Q(X)))\frac{1}{(2\pi^{n/2})|\Sigma|^{1/2}}$.*

Additionally, not only is the cost function of OP-DMA a lower bound on the c-Wasserstein divergence, but also itself is a Wasserstein divergence (the $c'$-Wasserstein divergence):

**Theorem 2.** *Let $W_c$ be a Wasserstein divergence. Then $W_{c'}$ is a Wasserstein divergence, with c' the prior-weighted c.*

As stated previously, our ideal mapping would place all inliers within regions where the probability was greater than some value, and all outliers into some regions where the probability is less than that value. Theorem 3 shows that this scenario is the optimal solution for the loss function of OP-DMA.

**Theorem 3.** *Let $Q$ be an encoder such that $\mathcal{D}(P_Q, P_Z, \mathcal{F}) = 0$, where $\mathcal{D}(A, B, \mathcal{F})$ is the Maximum Mean Discrepancy between A an B, $\mathcal{F}$ is the set of unbounded continuous functions and $P_Z = \mathcal{N}(\mathbf{0}, \mathbf{\Sigma})$. Let $X : \Omega \to \mathbb{R}^n$ be a centered random variable, $X \sim P_X$. Let $X(A)$, $A \subset \Omega$, be outliers and let $H = \Omega - A$ be the inliers, where $\int_{X(A)} p_X(x)dx = \alpha$. Furthermore, let $c'(a, G(Q(a)) > c'(h, G(Q(h)) \, \forall \, a \in X(A), h \in X(H)$. Then, the optimal solution of OP-DMA is to map such that $\|Q(X(A))\|_m \geq \delta$ and $\|Q(X(H))\|_m < \delta$, where $\delta = \sqrt{\int_0^{1-\alpha} \frac{t^{-n/2-1}e^{\frac{1}{2t}}}{2^{\frac{n}{2}}\Gamma(\frac{n}{2})}dt}$ and $\|\cdot\|_m$ is the Mahalanobis distance.*

Thus, the optimal solution for OP-DMA's cost function is one that maps outliers to regions with a larger Mahalanobis distance than that of inliers. This has the important implication that after

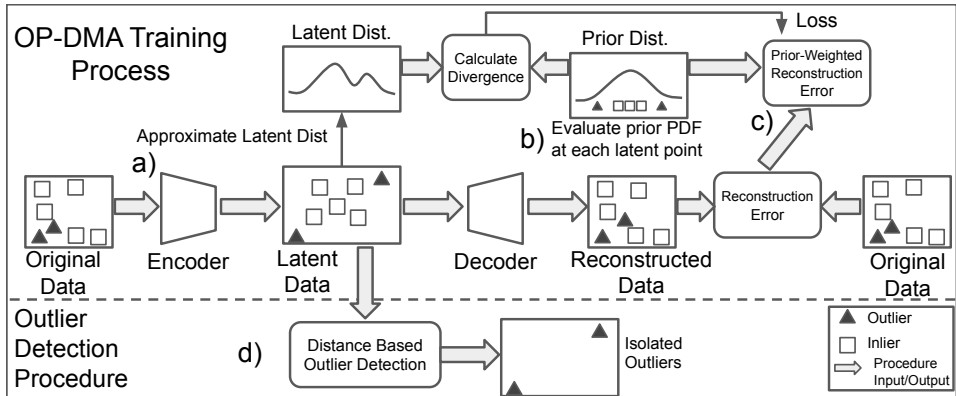

Figure 3: An overview of OP-DMA. a) Input data is encoded into latent space. b) The divergence between the latent distribution $P_Q$ and the prior distribution $P_Z$ is calculated and added to the loss term. c) The latent data is reconstructed, and the reconstruction error of each datapoint is weighted by the likelihood of observing the latent point according to the prior distribution $P_Z$. d) After training the network, outlier detection is performed by calculating the Mahalanobis distance of each point in the latent space

transformation with OP-DMA, outliers can be separated from inliers with a simple distance metric. This motivates out outlier detection scheme outlined in Section 3.3, where due to Theorem 1 we use the EllipticEnvelope method to identify outliers.

## 3.3 UNSUPERVISED OUTLIER DETECTION WITH OP-DMA

Lastly, we describe our end-to-end solution of outlier detection using OP-DMA. First, we transform the distribution of the dataset X to match a Gaussian distribution using OP-DMA. That is, the input dataset is transformed to match a prior distribution in the latent space of our autoencoder by a divergence term in the network's loss function that penalizes the divergence between the latent distribution and the prior distribution. For the purpose of outlier detection, we chose this prior to be a multivariate Gaussian distribution with 0 mean and identity covariance. Then, EllipticEnvelope (7) is used to determine outliers by fitting a Gaussian to the transformed data. Theorem 3 tells us that EllipticEnvelope is an appropriate choice for the outlier detection step, as it determines outliers based on Mahalanobis distance. However, other outlier detection methods such as IsolationForest (3) or OneClassSVM (4) could equally be leveraged. We use the empirical rather than the theoretical mean and covariance for the EllipticEnvelope as it is unrealistic to expect any such deep distribution mapping model to exactly match the prior. The outlier detection process is outlined in Algorithm 1 and in Figure 2. As seen in Algorithm 1, we use the unbiased estimator of the MMD from Gretton et. al. (22). For the kernel $k$ of the MMD we use the inverse multiquadratics kernel as is used in (16), and *Mean Squared Error* (MSE) for $c$. A high-level overview of the training and outlier detection process of OP-DMA is given in Figure 3.

## 4 EXPERIMENTAL EVALUATION

### 4.1 EXPERIMENTAL METHODOLOGY

**Synthetic Data Sets.** We determine possible failure cases for our method OP-DMA versus a standard autoencoder (AE). We use two synthetic datasets: (D1) Multimodal distribution in $\mathbb{R}^4$ with one cluster mean at (0,0,0,0) and another at (5,5,5,5) both with standard covariance and uniform noise added as outliers, and (D2) same multimodal distribution for inliers with a small cluster of outliers with a uniform probability in a small region between (-5,-5,-5,-5) and (-3,-3, -3, -3). In both cases, inliers account for 2.4% of the data.

**Benchmark Data Sets.** We compared OP-DMA to state-of-the-art methods from the ODDs database (8), a commonly used benchmark database of outlier detection datasets. Since we perform unsupervised outlier detection, we do not split each dataset into a training set and testing set. Instead,

---

**Algorithm 1:** Unsupervised Outlier Detection with OP-DMA

---

**Require :** Regularization coefficient $\lambda$
Contamination parameter $\alpha$
Initialized encoder network $Q_\Phi$ and decoder network $G_\Theta$ with random weights $\Phi$ and $\Theta$
Dataset $X$
**while** $\Theta$, $\Phi$ *not converged* **do**

 Sample $\{x_1, x_1, ..., x_n\}$ from $X$
 Sample $\{z_1, z_1, ..., z_n\}$ from $\mathcal{N}(\mathbf{0}, \mathcal{I})$
 Sample $\{\tilde{z}_1, \tilde{z}_1, ..., \tilde{z}_n\}$ from $Q_\Phi(Z|X)$
 Update weights $\Phi$ and $\Theta$ by descending

$$\frac{1}{n}\sum_{i=1}^n c(x_i, G_\Theta(\tilde{z}_i)) \cdot \lambda \cdot p_z(\tilde{z}_i) + \frac{1}{n(n-1)}\sum_{h \neq j} k(z_h, z_j) + \frac{1}{n(n-1)}\sum_{h \neq j} k(\tilde{z}_h, \tilde{z}_j) - \frac{2}{n^2}\sum_{h,j} k(z_h, \tilde{z}_j)$$

**end**
Find $D_{min} = \{Q_\Phi(x_i), Q_\Phi(x_j), ..., Q_\Phi(x_k)\}$, $\|D_{min}\| = (1-\alpha)\|D\|$ with Minimum
 Covariance Determinant estimator, $\inf_{\tilde{\Sigma}} Det\{\tilde{\Sigma}\}$.
Find estimated mean $\tilde{\mu}$ from $D_{min}$
**return** $\|Q_\Phi(x_i)\|_m = (Q_\Phi(x_i) - \tilde{\mu})'\tilde{\Sigma}(Q_\Phi(x_i) - \tilde{\mu})$ for $x_i \in D$ as outlier scores

---

outlier detection is performed in an unsupervised manner on the entire dataset. In each dataset all points are labeled as either inlier or outlier. We use these labels to calculate the weighted F1 score, but no method we test is trained on the labels. The datasets were chosen such that we could test our methods on datasets with a wide range of dimensionality in the feature space and with a wide range of outlier contamination percentage. Table 3 (in Appendix B) breaks doen the number of featurs, datapoints, and percentage of outliers in each dataset.

**Alternative Methods.** In order to determine the validity of our OP-DMA, we compare it to several state-of-the-art distribution mapping outlier detection methods. These include methods that perform outlier detection on the latent space of a *Wasserstein*, a *Variational Autoencoder*, and an *Adversarial Autoencoder* (25) all with a Gaussian prior, here denoted as WAE, VAE, and AAE respectively. Additionally, we test against MO-GAAL (26) and ALOCC (27), two state-of-the-art deep generative outlier detection models. We also test against LOF (23) and OC-SVM (4), two state-of-the-art non-deep learning outlier detection method.

**Parameter Configurations of Methods.** The encoders and decoders of all methods consist of 3-layer neural networks, where the decoder in each encoder-decoder pair mirrors the structure of the encoder. The number of nodes in the hidden layer of each network is a hyperparameter chosen from {5, 6, 9, 15, 18, 100}. The number of nodes in the output/latent layer is chosen from {2, 3, 6, 9, 15}. The value of the regularization parameter $\lambda$ is chosen such that the reconstruction error is on the same order of magnitude as the MMD error for the first epoch. We use the standard parameters of MO-GAAL from the authors' publically released code. Likewise, we use the standard configuration of ALOC from the authors code, although we add an additional dense layer at the beginning of each subnetwork. We do this as ALOCC is a convolutional method and assumes input to be images of a certain shape. The additional dense layer we add to the model transforms the inut data from its original dimensionality into the required shape. We use the standard parameters from Scikit-Learn for LOF and OC-SVM.

## 4.2 EXPERIMENTAL RESULTS

**Synthetic Experiments.** Figure 4 A shows the ratio of the recreation error for outliers/inliers for OP-DMA versus an autoencoder (AE) for each synthetic dataset. For uniform noise (D1), both OP-DMA and AE feature a high ratio for the error of outlier/inliers. In this case, a standard AE would be adequate to find the outliers. However, for data where outliers are a small cluster (D2), the ratio between outliers to inliers goes quickly to 1 for AE whereas it levels off to ~10 for OP-DMA. In this case, our OP-DMA succeeds to identify the outliers, while AE cannot. We also verify that OP-DMA maps outliers to low-probability regions of the latent space. Figure 4 B shows the latent space of

OP-DMA for the uniform outlier dataset. The outliers occur far from the mean in low-probability regions. The average probability in the prior for outliers after transformation was 0.02, whereas the average probability of the inliers according to the prior was 0.08

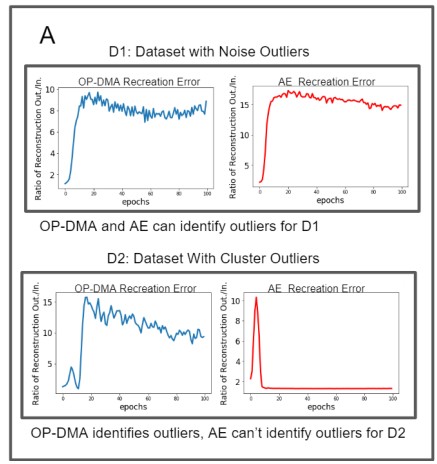
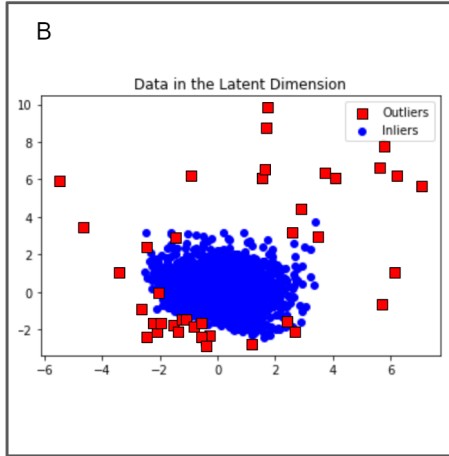

Figure 4: A: Reconstruction error vs. epoch for OP-DMA and a standard autoencoder for both noise outliers and cluster outliers. B: Latent representation of data in OP-DMA for uniform outliers.

**Outlier Detection Accuracy.** The results reported in Table 2 are the maximum weighted F1-scores of each method over all contamination rates from 0.1 to 1.0 with a step size of 0.1.

| | OP-DMA | WAE | VAE | AAE |
|---|---|---|---|---|
| **Running Time (seconds)** | 827.14 | 723.51 | 857.46 | 971.51 |

Table 1: Running time for methods tested

**Running Time.** In order to determine the feasibility of using OP-DMA instead of one of the leading methods in practice, we ran an experiment to determine the running times of each method. This consisted of performing 5 runs of OP-DMA, WAE, VAE, and AAE on the Satellite dataset (24). The average running time of each method over the 5 iterations is shown in the following table. Each method was run on a machine with 32GB of DDR4 RAM and a Intel Xeon Platinum 8160 CPU.

## 5 DISCUSSION

OP-DMA outperforms the state-of-the-art methods on the majority of the outlier detection datasets. This is because unlike the state-of-the-art methods, OP-DMA actively encourages outliers to be mapped to low probability regions instead of relying on outliers to be poorly mapped to the prior. While in this work, we focused on unsupervised outlier detection, OP-DMA could be extended to *supervised* outlier detection by making the following change to the objective function:

$$L = \inf_{Q:P_Q=P_Z} \mathbb{E}_{P_{X_{in}}} \mathbb{E}_{Q(Z|X)}[c(X_{in}, G(Q(Q(X_{in}))))]$$
$$+ \mathbb{E}_{P_{X_{out}}} \mathbb{E}_{Q(Z|X)}[c(X_{in}, G(Q(Q(X_{out})))) \cdot p_Z(Q(X_{out}))] + \lambda \mathcal{D}(P_Q, P_Z),$$

where $X_{in}$ corresponds to the set of labeled inlier points and $X_{out}$ is the set of outliers. OP-DMA can also be modified for *semisupervised* OP-DMA by pretraining on labeled inliers without weighting the reconstruction error by the likelihood, and then performing OP-DMA as is done in the unsupervised version on the remaining unlabeled datapoints.

## 6 CONCLUSION

We have introduced OP-DMA, an autoencoder-based distribution mapping method for outlier detection that maps outliers in the feature space to low probability regions in the latent space in which a multivariate standard normal Gaussian prior distribution is enforced. Outliers are consequently

| Dataset | OP-DMA | WAE | VAE | AAE | MO-GAAL | ALOCC | LOF | OC-SVM |
|---------|--------|-----|-----|-----|---------|-------|-----|--------|
| Satellite | **0.735** ±0.012 | 0.554 ±0.009 | 0.310 ±0.007 | 0.480 ±0.008 | 0.481 ±0.001 | 0.706 ±0.008 | 0.413 | 0.417 |
| Pima | **0.625** ±0.018 | 0.520 ±0.020 | 0.23 ±0.019 | 0.497 ±0.007 | 0.518 ±0.002 | 0.517 ±0.010 | 0.456 | 0.440 |
| WBC | **0.590** ±0.011 | 0.448 ±0.013 | 0.268 ±0.011 | 0.19 ±0.018 | 0.468 ±0.060 | 0.529 ±0.007 | 0.480 | 0.199 |
| Arrythmia | 0.531 ±0.017 | **0.601** ±0.015 | 0.201 ±0.010 | 0.294 ±0.010 | 0.518 ±0.011 | 0.457 ±0.020 | 0.464 | 0.254 |
| Breastw | **0.951** ±0.014 | 0.950 ±0.011 | 0.368 ±0.009 | 0.479 ±0.007 | 0.944 ±0.014 | 0.863 ±0.062 | 0.292 | 0.824 |
| Letter | 0.182 ±0.001 | 0.091 ±0.003 | 0.048 ±0.005 | 0.10 ±0.002 | 0.159 ±0.003 | 0.165 ±0.001 | **0.488** | 0.208 |
| Cardio | **0.590** ±0.013 | 0.290 ±0.012 | 0.221 ±0.008 | 0.204 ±0.009 | 0.542 ±0.032 | 0.432 ±0.071 | 0.208 | 0.323 |
| Lympho | 0.585 ±0.012 | 0.443 ±0.008 | 0.341 ±0.011 | 0.310 ±0.018 | 0.719 ±0.119 | **0.958** ±0.083 | 0.833 | 0.150 |
| Musk | 0.32 ±0.007 | 0.330 ±0.010 | 0.243 ±0.009 | 0.228 ±0.025 | **0.394** ±0.090 | 0.199 ±0.160 | 0.069 | 0.101 |
| Thyroid | **0.29** ±0.019 | 0.173 ±0.021 | 0.130 ±0.019 | 0.170 ±0.023 | 0.225 ±0.027 | 0.084 ±0.026 | 0.200 | 0.090 |
| Satimage-2 | **0.860** ±0.039 | 0.176 ±0.013 | 0.148 ±0.007 | 0.535 ±0.015 | 0.814 ±0.049 | 0.818 ±0.026 | 0.122 | 0.019 |

Table 2: Weighted F1 scores for OP-DMA vs state-of-the-art methods on datasets from the ODDs outlier detection database with 95% confidence interval.

easily identifiable in the latent space. Our experimental study comparing OP-DMA to state-of-the-art methods on a collection of different benchmark outlier detection datasets shows that it outperforms WAE, VAE, and AAE on the majority of the datasets. We have also demonstrated that there is not a significant increase in running time between our method and state-of-the-art methods.

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

# Appendices

## A
## PROOFS

**Theorem 1.** *Let $P_Z = \mathcal{N}(\mu, \Sigma)$. Then $W_{c'}(X, G(Q(X))) \leq W_c(X, G(Q(X))) \frac{1}{(2\pi^{n/2})|\Sigma|^{1/2}}$. Additionally, if $|\Sigma| \leq \frac{1}{(4\pi^n)}$ then $W_{c'}^\lambda(X, G(Q(X))) \leq W_c^\lambda(X, G(Q(X))) \frac{1}{(2\pi^{n/2})|\Sigma|^{1/2}}$.*

*Proof.* We will first show that $W_{c'}(X, G(Q(X))) \leq W_c(X, G(Q(X))) \frac{1}{(2\pi^{n/2})|\Sigma|^{1/2}}$, and then show that $W_{c'}^\lambda(X, G(Q(X))) \leq W_c^\lambda(X, G(Q(X))) \frac{1}{(2\pi^{n/2})|\Sigma|^{1/2}}$.

The maximum value of $p_Z(z)$ will occur at $z = \mu$. Then, $\sup p_Z = \frac{1}{(2\pi^{n/2})|\Sigma|^{1/2}}$. Thus

$$W_{c'}(X, G(Q(X))) = \inf_{Q:P_Q=P_Z} \mathbb{E}_{P_X} \mathbb{E}_{Q(Z|X)} [c(X, G(Q(X))) \cdot p_Z(Q(X))]$$

$$\leq \left( \frac{1}{(2\pi^{n/2})|\Sigma|^{1/2}} \right) \left( \inf_{Q:P_Q=P_Z} \mathbb{E}_{P_X} \mathbb{E}_{Q(Z|X)} [c(X, G(Q(X)))] \right)$$

$$= \frac{1}{(2\pi^{n/2})|\Sigma|^{1/2}} W_C(X, G(Q(X)))$$

Similarly, to show that the prior-weighted c-Wasserstein divergence is a constant multiple of the true Wasserstein divergence:

$$W_{c'}^\lambda(X, G(Q(X))) = \inf_{Q:P_Q=P_Z} \mathbb{E}_{P_X} \mathbb{E}_{Q(Z|X)} [c(X, G(Q(X))) \cdot p_Z(Q(X))] + \lambda \mathcal{D}(P_Q, P_Z)$$

$$\leq \left( \frac{1}{(2\pi^{n/2})|\Sigma|^{1/2}} \right) \left( \inf_{Q:P_Q=P_Z} \mathbb{E}_{P_X} \mathbb{E}_{Q(Z|X)} [c(X, G(Q(X)))] + \lambda \mathcal{D}(P_Q, P_Z) \right)$$

$$= \frac{1}{(2\pi^{n/2})|\Sigma|^{1/2}} W_C^\lambda(X, G(Q(X)))$$

$\square$

**Theorem 2.** *Let $W_c$ be a Wasserstein divergence. Then $W_{c'}$ is a Wasserstein divergence, with c' the prior-weighted c.*

*Proof.* Since $c$ is a Wasserstein divergence, we know that $c(x_1, x_2) \geq 0 \; \forall \; x_1, x_2 \in supp(P)$, $c(x, x) = 0 \; \forall \; x \in supp(P)$, and $\mathbb{E}_\gamma[c(x_1, x_2)] \geq 0 \; \forall \; \gamma \in \Gamma[P, P_z]$. Since $P_Z(z) \geq 0 \; \forall \; z$, $c'$ will also fulfill the three aforementioned properties of $c$. Thus, $W_{c'}$ is a Wasserstein divergence. $\square$

**Theorem 3.** *Let $Q$ be an encoder such that $\mathcal{D}(P_Q, P_Z, \mathcal{F}) = 0$, where $\mathcal{D}(A, B, \mathcal{F})$ is the Maximum Mean Discrepancy between A an B, $\mathcal{F}$ is the set of unbounded continuous functions and $P_Z = \mathcal{N}(\mathbf{0}, \Sigma)$. Let $X : \Omega \to \mathbb{R}^n$ be a centered random variable, $X \sim P_X$. Let $X(A)$, $A \subset \Omega$, be outliers and let $H = \Omega - A$ be the inliers, where $\int_{X(A)} p_X(x) dx = \alpha$. Furthermore, let $c'(a, G(Q(a)) > c'(h, G(Q(h)) \; \forall \; a \in X(A), h \in X(H)$. Then, the optimal solution of OP-DMA is to map such that $\|Q(X(A))\|_m \geq \delta$ and $\|Q(X(H))\|_m < \delta$, where $\delta = \sqrt{\int_0^{1-\alpha} \frac{t^{-n/2-1} e^{\frac{1}{2t}}}{2^{\frac{n}{2}} \Gamma(\frac{n}{2})} dt}$ and $\| \cdot \|_m$ is the Mahalanobis distance.*

*Proof.* The Mahalanobis distance of $Q(X)$ can itself be expressed as a random variable, $\delta = \sqrt{Q(X)\Sigma^{-1}Q(X)^T}$. Let $\Phi_\delta$ be the CDF of $\delta$. Then, $\Phi_\delta(1 - \alpha) = P(\delta \leq 1 - \alpha) = P(\delta^2 \leq (1-\alpha)^2) = \Phi_{\delta^2}((1-\alpha)^2)$.

Let $Y = Q(X)M^{-1}$, where $M^T M = \Sigma$ is the Choleski decomposition of the covariance $\Sigma$. Since $\mathcal{D}(P_Q, P_Z, \mathcal{F}) = 0$, and $\mathcal{D}(A, B, \mathcal{F}) = 0$ iff $A = B$, we thus know that $Q(X) \sim \mathcal{N}(\mathbf{0}, \Sigma)$. Thus since

$Q(X)$ is normally distributed and centered, $Y$ thus is normally distributed with identity covariance. Since $\delta^2 = Q(X)\Sigma^{-1}Q(X)^T = YY^T$, $\Phi_{\delta^2}$ is the CDF of of the sum of squares of $n$ normally distributed variables with mean 0 and $\sigma = 1$. Thus, $\Phi_{\delta^2}$ is the Chi Squared distribution. The inverse Chi Squared CDF will therefore give us the distance $\delta$ such that $1 - \alpha$ percent of the points are within $\delta$:

$$\delta = \sqrt{\int_0^{1-\alpha} \frac{t^{-n/2-1}e^{\frac{1}{2t}}}{2^{\frac{n}{2}}\Gamma(\frac{n}{2})}dt}$$

Now, let us assume that for some parameter choice $\Theta'$ for $Q$ that $\alpha P(Q(X(A)|\Theta') \leq \delta) = \beta$, $\beta > 0$. Consequently, $(1 - \alpha)P(Q(X(H)|\Theta') > \delta) = \beta$, since $P(Q(X) > \delta) = \alpha$ and $\int_{X(A)} p_X(x)dx = \alpha$. Conversely, let us assume that there is a parameter configuration $\Theta$ such that $\alpha P(Q(X(A)|\Theta) \leq \delta) = 0$ and so $(1 - \alpha)P(Q(X(H)|\Theta) > \delta) = 0$.

Since $P_Z \sim \mathcal{N}(\mathbf{0}, \Sigma)$, $p_Z(d_1) < p_Z(d_2)$ for $\|d_1\|_m > \|d_2\|_m$. Thus, since we assume $c(a, G(Q(a)) > c(h, G(Q(h)) \,\forall\, a \in X(A)$, $h \in X(H)$, then $\mathbb{E}_{P_X}\mathbb{E}_{Q(Z|X)}c'(x_p, G(Q(x_p|\Theta'))) = \mathbb{E}_{P_X}\mathbb{E}_{Q(Z|X)}c(x_p, G(Q(x_p|\Theta')))p_z(x_p) > \mathbb{E}_{P_X}\mathbb{E}_{Q(Z|X)}c(x_p, G(Q(x_p|\Theta)))p_z(x_p) = \mathbb{E}_{P_X}\mathbb{E}_{Q(Z|X)}c'(x_p, G(Q(x_p|\Theta)))$. Thus, a mapping that places outliers $\delta$ away from the mean will have a lower loss than one that places some percentage of outliers less than $\delta$ from the mean. $\qquad\square$

# B
## DATASETS

| Dataset | # Features | # Datapoints | % Outliers |
|---|---|---|---|
| Satellite | 36 | 6435 | 32% |
| Pima | 8 | 768 | 35% |
| WBC | 30 | 278 | 5.6% |
| Arrythmia | 274 | 452 | 15% |
| Breastw | 9 | 638 | 35% |
| Letter | 32 | 1600 | 6.25% |
| Cardio | 21 | 1831 | 9.6% |
| Lympho | 18 | 148 | 4.1% |
| Musk | 166 | 3062 | 3.2% |
| Thyroid | 6 | 3772 | 2.5% |
| Satimage-2 | 36 | 5803 | 1.2% |

Table 3: Description of the dimensionality, number of datapoints, and contamination percentage of each dataset we tested our method on.

# C
## SENSITIVITY TO CONTAMINATION PARAMETER

While the contamination parameter $\alpha$ is not used during the training of OP-DMA, it is used to fit the Elliptic Envelope on the encoded data after training. We test the sensitivity of this method to the value of the contamination parameter by evaluating the F1 score of outlier detection on the Satellite dataset after encoding the data with OP-DMA. The results are shown in the figure below.

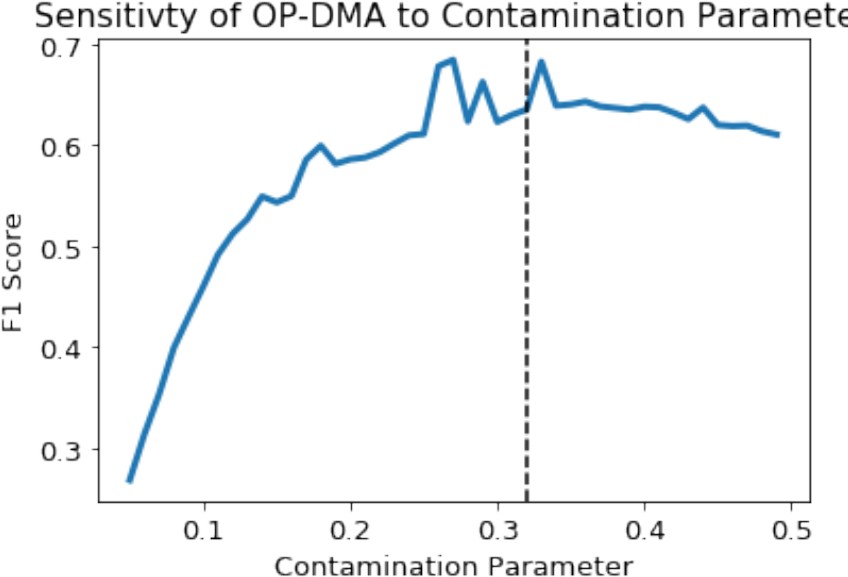

Figure 5: The F1 score after outlier detection using OP-DMA + Elliptic Envelope on the Satellite dataset, for various values of the contamination parameter $\alpha$. The vertical dashed line shows the true percentage of outliers in the dataset.

This analysis shows that as long as the contamination parameter is not significantly less than the true contamination percentage the F1-score is robust to different values of the contamination parameter.

