# OpenReview forum: "Versatile Anomaly Detection with Outlier Preserving Distribution Mapping Autoencoders"
_ICLR.cc/2020/Conference — Reject_

### Official Review · AnonReviewer3 · 2019-10-17
**Official Blind Review #3**

**Rating:** 6

**Review:**

This paper proposes a novel outlier detection approach, based on Wasserstein auto encoders.

Unfortunately, I cannot comment on the overall scientific contribution of the paper, as I simply do not possess the expertise to judge it accurately. I will rely on the judgement of the other reviewers, whom I hope will have more experience and will better know the literature. I will report on a few issues with aspects related to the presentation below.

In fig. 1, the (a) and (b) should probaby appear below each diagram. "on trained" is repeated twice in the caption.

In fig. 2, the WAE acronym is defined only much later in the text.

Fig. 2 is also a bit confusing, since it seems like the order in which the diagrams appear should be swapped. Indeed, the text also refers to fig. 2 (b) before (a). The text just below fig. 2 also refers to Figure 1, but I think it should be 2?

In sec. 4.2, the text mentions table 2 when it should really be table 1. Also, table 1 should appear before table 2 in the body.

It looks as if the symbols () and [] are inverted? All references are referred to with () and text within parentheses (e.g. references to figures) have [].


**Experience Assessment:**

I do not know much about this area.

**Review Assessment: Checking Correctness Of Derivations And Theory:**

I did not assess the derivations or theory.

**Review Assessment: Checking Correctness Of Experiments:**

I assessed the sensibility of the experiments.

**Review Assessment: Thoroughness In Paper Reading:**

I made a quick assessment of this paper.

---

> ### Author Response · Authors · 2019-11-13
> **Thank you**
>
> OUR RESPONSE. Thank you for your time spent reviewing our paper. We have fixed the typos and incorrect figure/table references that have have pointed out in your review.
>
> QUESTION 1: “Fig. 2 is also a bit confusing, since it seems like the order in which the diagrams appear should be swapped. Indeed, the text also refers to fig. 2 (b) before (a). The text just below fig. 2 also refers to Figure 1, but I think it should be 2? “
>
> OUR RESPONSE. Thank you for alerting us to this mistake. We have corrected the references to the figure in the paper, so that the text now refers to the correct subfigure.

---

### Official Review · AnonReviewer5 · 2019-11-03
**Official Blind Review #5**

**Rating:** 6

**Review:**

This work proposes an outlier detection method based on WAE framework. WAE is trained to ensure that 1) latent distribution follows a prior distribution 2) weighted reconstruction error is low where prior PDF is used to weight the reconstruction error.

Positives
------------
1.I liked the intuition behind the proposed method and I felt its worth exploring. Paper points out that in previous works, there is no mechanism to prevent outliers from getting mapped to high probability areas in the model [21]. Authors claim that their method will over-come this issue. I believe this is the core contribution of the paper.

2.I agree with authors point that WAE is a better choice than VAE for outlier detection because, former "encourages the latent representations as a whole to match the prior ".

3.I agree training with a distributional divergence loss along with a weighed reconstruction loss will be helpful for learning a robust representation (as outliers in the training dataset would be assigned a lower weight).

4.Authors have compared the performance of their method on OOD dataset where they have  compared against three other baseline methods, where they obtain better performance in majority of cases.


Negatives
--------------
A. Outlier detection and anomaly/novelty detection are two very different problems.  Outliers are 'bad eggs' coming from the same class as normal data. On the other hand, anomaly/novelty are unexpected data possibly coming from other classes. This is the taxonomy followed by majority of works. In my understanding this work is about 'outlier detection'. I hope authors will use the term 'outlier detection' consistency through the paper.

B. Authors have not done a good survey on existing outlier detection methods.  Eg:
    I. Chong You, Rene Vidal, Provable Self-Representation Based Outlier Detection in a Union of Subspaces, CVPR 17
    II. Yan Xia, Xudong Cao, Fang Wen, Gang Hua, Jian Sun, Learning Discriminative Reconstructions for Unsupervised Outlier Removal, ICCV 15
   III. Mohammad Sabokrou, Mohammad Khalooei, Mahmood Fathy, Ehsan Adeli, Adversarially Learned One-Class Classifier for Novelty Detection, CVPR 18. (they have experiments on outlier detection)

C. Although existing methods have not explicitly stated the problem identified in (1) above, their proposals are indirectly solving this problem. Therefore, authors should have compared with papers listed in (B) to demonstrate the effectiveness of their method for a meaningful comparison.

D. This is the first time I'm seeing the OOD dataset used in the paper. Have other works published their results on this dataset? Can they be included in your paper?. If not, consider reporting results on standard datasets used in papers (B). I believe reporting results on at least two datasets is necessary to demonstrate the generalizability of the method.

Other comments
------------------------
a. What is the dimensionality used in the latent space? I believe a larger latent space may be required in modeling more complex data such as images. Is the weighting mechanism effective when a very large latent space is used due to the curse of dimensionality.

b. I don't think the synthetic dataset experiment is giving any interesting insights. This space is better used if an additional dataset is used instead.

In conclusion, I like the idea presented in this paper; however, I believe experimental results needs to be improved significantly to demonstrate the effectiveness of the proposed method. I cannot recommend to accept the paper in its present condition.


Post Rebuttal:
Authors have partially addressed my concerns. In light of new experiments provided, I'm changing my decision to weak accept.

**Experience Assessment:**

I have published one or two papers in this area.

**Review Assessment: Checking Correctness Of Derivations And Theory:**

I assessed the sensibility of the derivations and theory.

**Review Assessment: Checking Correctness Of Experiments:**

I carefully checked the experiments.

**Review Assessment: Thoroughness In Paper Reading:**

I read the paper thoroughly.

---

> ### Author Response · Authors · 2019-11-14
> **Thank you**
>
> Thank you for your feedback and insight from your experience in this domain.  Unfortunately, we do not believe we can edit the name of the paper on this submission site in order to remove the reference to "anomaly" detection. However, all references to "anomalies" have been changed to "outliers" within the actual document. To address your other points:
>
> QUESTION 1:  “Authors have not done a good survey on existing outlier detection methods.  Eg:...”
>
> OUR RESPONSE.   Thank you for alerting us about these particular methods. As you have suggested, we will add these three papers to our related work section.
>
> We have added the method used “Adversarially Learned One-Class Classifier for Novelty Detection” (ALOCC) as one of the methods we compare our OP-DMA against. While the authors have released their implementation of ALOCC, this method was designed to work with image data. As we did not wish to drastically change the architecture of this method in order to work with non-image data, as added a single dense layer to the beginning of the network that takes in input data of any shape and outputs the data in a shape that can be accepted by the following convolutional layer. Our analysis shows that we outperform ALOC in all but 1 of the dataset we tested.
>
> Additionally, we have now  added into our comparative study an additional recent state-of-the-art deep outlier detection method [1]. We have chosen this method not only because it is also a deep neural network method, but also in addition because this work had in fact  already compared their method against several datasets we also have evaluated OP-DMA against.
>
> As you can see in our experimental result table in Table 2, OP-DMA outperforms this state-of-the-art method on all ODDs datasets besides 2 datasets.
>
>
> QUESTION 2:  “This is the first time I'm seeing the OOD dataset used in the paper. Have other works published their results on this dataset? Can they be included in your paper?. If not, consider reporting results on standard datasets used in papers (B). I believe reporting results on at least two datasets is necessary to demonstrate the generalizability of the method”
>
> OUR RESPONSE. We believe that there may be some misunderstanding here. Namely, OOD is not a data set, but instead a benchmark repository that archives and makes available a rich variety of distinct  (labeled) data sets for outlier analysis research.  It is indeed a popular benchmark used by related work systems focussing on outlier research in their experimental studies.
> Further, we note that we have tested on 11 different datasets all coming from this repository, not just 1. While all of the datasets came from the ODD repository, each of the 11 datasets we worked with are unique real-world data sets. Also, as stated above, we have added in a comparison to this select recent deep outlier detection method [1] which had already been evaluated on several of the datasets we used also from this ODD repository.
>
> [1] Generative Adversarial Active Learning for Unsupervised Outlier Detection
> Yezheng Liu, Zhe Li, Chong Zhou, Yuanchun Jiang, Jianshan Sun, Meng Wang & Xiangnan He
> IEEE Transactions on Knowledge and Data Engineering (TKDE 2019)

---

### Official Review · AnonReviewer4 · 2019-11-04
**Official Blind Review #4**

**Rating:** 3

**Review:**

The paper proposes an improved extension of the Wasserstein auto-encoder for anomaly detection. The novelty is in proposing a weighted reconstruction error that penalizes the mapping of data with high reconstruction errors (mostly anomalies) into high probability regions. The idea being that an outlier would have a higher reconstruction error, and hence should be mapped to low-probability region of the latent distribution.

Experimental Results:  As a distribution mapping auto-encoder model, OP-DMA outperforms the deep learning based state-of-the-art models in the same domain.

Overall Assessment: The authors have a nice idea of forcing the latent mappings of inputs to correlate with their reconstruction error. Overall, the method is promising, but I have the following concerns:

* Using the reconstruction error as an anomaly score has been explored many years ago (check replicator neural networks), the novelty here is to enforce that on the latent space in the context of a variational auto-encoder. I am not sure if, from anomaly detection perspective, this is any better than simply using the reconstruction score. Why go the VAE route at all?
* Is there a possibility that assuming a single multi-variate Gaussian, as a prior, too restrictive? Could it result in a high false alarm rate as well? I guess this could be answered by more experimental results on richer data sets (even synthetic is fine).

* In most score based algorithms, the anomaly score is computed without assuming any prior knowledge about the contamination proportion. However, in the case of OP-DMA, the contamination parameter is used to train the auto-encoder that scores the data. This might result in an optimization that is very specific to the parameter setting. I strongly recommend a sensitivity analysis to study the robustness of the model against different values of contamination parameter.

* Performance on synthetic data-set has not been presented. The set H in theorem 3 has not been defined.

* Additional comparison with other non-distribution mapping state-of-the-art models such as LOF, oc-SVM, KNN would give a clearer idea of the performance. This is important, because in my past experience, non-deep learning methods give much better results on the benchmark data sets that the authors have evaluated their method on. In fact, a comparative analysis (See - https://journals.plos.org/plosone/article?id=10.1371/journal.pone.0152173) gives a very nice comparison. However, since the authors provide results using Avg F1-score, instead of AUC curve, it was not possible to compare them myself.

* In figure 3, in the training process, the authors have describe to add the divergence between the latent and prior distribution to the loss function, however, nothing like this is clearly shown in the figure.The references of figures in the text are either out of place or incorrect. Figure 1(a) and (b) in reference to the text are incorrect. Figure 2 is the misleading figure as it doesn't illustrate the anomaly detection process. Figure 3 has not been mentioned anywhere in the text. The authors have mentioned the comparison of their method with Wasserstien and variational auto-encoders in the text, while in table 2 and 4, AAE has also been shown as one of the method for comparison, which is never mentioned or described in the text.

* Typo in caption of Figure 1 and the first line of section 3.3

Overall, I am hesitant to recommend the paper before cross-checking the issue with contamination proportion and learning more about how a VAE framework is indeed important for anomaly detection.


**Experience Assessment:**

I have published in this field for several years.

**Review Assessment: Checking Correctness Of Derivations And Theory:**

I assessed the sensibility of the derivations and theory.

**Review Assessment: Checking Correctness Of Experiments:**

I carefully checked the experiments.

**Review Assessment: Thoroughness In Paper Reading:**

I read the paper at least twice and used my best judgement in assessing the paper.

---

> ### Author Response · Authors · 2019-11-13
> **Thank you**
>
> We thank you for your detailed feedback and time on our work. Your insights are very much appreciated. We have fixed the typos, incorrect references to figures and added in reference to AAEs in our experimental section.
>
> QUESTION  1.  “the novelty here is to enforce that on the latent space in the context of a variational auto-encoder. I am not sure if, from anomaly detection perspective, this is any better than simply using the reconstruction score. Why go the VAE route at all?”
>
> OUR RESPONSE. To address your question, the reason the distribution mapping framework is important is that it allows us to calculate the likelihood of each point in the latent space, which can be leveraged to weight the reconstruction error of each point by its likelihood. The reason we do this weighting  rather than just directly using this reconstruction error is that in standard autoencoders the average reconstruction error for outliers and the average reconstruction error for inliers often tends to converge to the same value. This is in fact what we show experimentally in Figure 4 (a).
>
> As can be seen, the autoencoder initially has a higher reconstruction error for outliers than inliers, but iit is quickly able to reproduce both inliers and outliers roughly equally well before converging. Our OP-DMA solution, on the other hand, succeeds  to maintain this difference in reconstruction error throughout the training process. We accomplish this by weighting the reconstruction loss for outliers by their likelihood in the latent space. By performing distribution mapping and making the distribution of the latent space match a prior distribution for which the PDF is known and tractible, we can calculate the likelihood of each point in the latent space.
>
> Also, just to be clear,  we indeed use a WAE architecture  rather than a VAE solution in our work as you can see in Section 3. The reason for this is that, unlike a VAE, the WAE encourages the latent representations as a whole to match the prior, as also detailed in our Section 3.
>
> QUESTION 2:   “Is there a possibility that assuming a single multi-variate Gaussian, as a prior, is too restrictive? Could it result in a high false alarm rate as well?”
>
> OUR RESPONSE. A multivariate Gaussian prior for the latent space of VAEs and WAEs has been shown to be sufficient to represent data in well in a variety of domains including those characterized by very complicated data, including  images [1] and text [2].
>
> QUESTION 3:   “In most score-based algorithms, the anomaly score is computed without assuming any prior knowledge about the contamination proportion. However, in the case of OP-DMA, the contamination parameter is used to train the auto-encoder that scores the data.”
>
> OUR RESPONSE.  No, this observation is incorrect.
> The autoencoder of OP-DMA does ***NOT***  use the contamination parameter. Only the EllipticEnvelope method requires the contamination parameter, in order to fit a robust covariance estimate to the encoded data. This means that one could switch out another algorithm for the anomaly detection step, such as OC-SVM, that does not require the contamination parameter and nothing about the OP-DMA network training procedure would change.
>
> Still, to address your concern, we have now added an additional sensitivity analysis of OP-DMA on the Satellite dataset to the Appendix of our paper. This analysis shows that the performance of EllipticEnvelope on the encoded data is robust as long as the contamination percentage is not grossly underestimated.
>
> QUESTION 4“Additional comparison with other non-distribution mapping state-of-the-art models such as LOF, oc-SVM, KNN would give a clearer idea of the performance.”
>
> OUR RESPONSE.  As you have requested, we have now added additional experiments with the state-of-the-art models including the ones you have suggested, namely, OC-SVM and LOF, into our paper. The results are shown in Table 2.  It is apparent from the results that indeed our  OP-DMA solution consistently outperforms these state-of-the-art methods in all but 2 of the datasets.
>
> QUESTION 5: “The set H in theorem 3 has not been defined”
>
> OUR RESPONSE: We are sorry for this mistake. The set H are the inlier points. We have added this to the statement of theorem 3.
>
> QUESTION 4: “In figure 3, in the training process, the authors have describe to add the divergence between the latent and prior distribution to the loss function, however, nothing like this is clearly shown in the figure.”
>
> OUR RESPONSE: Thank you for alerting us to this. We have added an arrow from the divergence to the loss function in Figure 3.
>
> [1] Pu, Yunchen, et al. "Variational autoencoder for deep learning of images, labels and captions." Advances in neural information processing systems. 2016.
>
> [2] Yang, Zichao, et al. "Improved variational autoencoders for text modeling using dilated convolutions." Proceedings of the 34th International Conference on Machine Learning-Volume 70. JMLR. org, 2017.

---

### Decision · Program_Chairs · 2019-12-19

**Decision:**

Reject

**Comment:**

This paper proposes an outlier detection method that maps outliers to low probability regions of the latent space. The novelty is in proposing a weighted reconstruction error penalizing the mapping of outliers into high probability regions. The reviewers find the idea promising.
They have also raised several questions. It seems the questions are at least partially addressed in the rebuttal, and as a result one of our expert reviewers (R5) has increased their score from WR to WA. But since we did not have a champion for this paper and its overall score is not high enough, I can only recommend a reject at this stage.